# The Development and Evaluation of a Training Monitoring System for Amateur Rugby Union

**Alan Griffin** [1,2,*], **Ian C. Kenny** [1,2], **Thomas M. Comyns** [1,2] **and Mark Lyons** [1,2]

1    Department of Physical Education and Sport Sciences, University of Limerick, V94 T9PX Limerick, Ireland; ian.kenny@ul.ie (I.C.K.); tom.comyns@ul.ie (T.M.C.); mark.lyons@ul.ie (M.L.)
2    Health Research Institute, University of Limerick, V94 T9PX Limerick, Ireland
*    Correspondence: alan.griffin@ul.ie; Tel.: +35-38-7770-2119

**Abstract:** A training monitoring system (TMS) should be both attainable and scientifically grounded; however, the optimal method of monitoring training is not yet fully understood. The purpose of this study was to develop and evaluate an online TMS for amateur rugby union. The experimental approach to the problem consisted of five phases: (1) establishing the current training and training load (TL) monitoring practices of amateur rugby union teams, (2) designing and developing the TMS, (3) recruiting teams and subsequently introducing the TMS, (4) supporting the strength and conditioning (S&C) coaches using the TMS, and (5) evaluating the TMS. The findings of this study support the use of an online TMS as a useful and effective method of facilitating training prescription and design in an effort to reduce injury risk and enhance performance. The main barriers impeding player compliance are the lack of feedback on their data and evidence of its use in training design, coaching, and prescription. The effectiveness of the system is dependent on the extent to which the associated challenges are mitigated to ensure quality and consistent data. However, this study offers a method of monitoring training that can be effective while also establishing pitfalls to avoid for both practitioners and researchers alike.

**Keywords:** training; monitoring; amateur; coaching; injury; performance; rugby

## 1. Introduction

Training inflicts stress on an athlete that could provoke undesired effects in terms of athlete wellbeing [1]. The monitoring of athletes' preparedness for, and response to, various training stimuli may provide practitioners with greater opportunity to prescribe and design training with the aim of maximising recovery and performance while simultaneously minimising risk of injury, illness, and, health and wellbeing problems [1–5]. Monitoring training over time can be employed to assess athletes' stress levels, fatigue, and readiness to train [6]. A successful training monitoring system (TMS) requires effective and consistent collection, analysis, and application of data that will allow adjustment of training prescription without overburdening the players [4,5]. This can be a particular challenge at the amateur level due to a lack of resources, time, and financial compensation received by the S&C coaches [7].

Training load (TL) can be defined as the cumulative amount of stress placed on an athlete from single or multiple training sessions and competitions over a period of time [5,8]. TL has a known association with injury, illness, recovery, and performance [7–12]. Monitoring TL is essential to effective load management, athlete adaptation, and injury mitigation in sport [5]. Additionally, there is evidence indicating that a disturbance in psychosocial stress in athletes is associated with an increased risk of injury and illness and can be effectively monitored using measures of wellbeing [4,13]. Subjective measures respond to training-induced changes in athlete wellbeing and may also be more

sensitive and reliable than objective measures [1,2,10]. Therefore, it is recommended that athletes report their subjective wellbeing on a regular basis alongside other athlete monitoring practices [1].

A TMS should be both attainable and scientifically grounded [14]; however, the optimal method of monitoring training or TL is not yet fully understood and further research is required to optimise training monitoring practices specifically [2,4,5,7]. The effectiveness of a TMS should be determined by getting the perspective of the end-users (i.e., coaches and players) and then factors influencing its implementation can be addressed [4,15].

Rugby union, hereafter rugby, is a high-intensity invasion game with approximately 9.6 million registered players worldwide, the vast majority of these playing at the amateur level [16]. Rugby has an inherent risk of injury due to its associated physical nature [17] and a recent systematic review found that amateur rugby has a match injury incidence rate of 46.8/1000 player hours [18]. This, combined with the known detrimental effects of injury on player welfare [19–22] and team and individual athletic performance [11,23], led to the development of the Irish Rugby Injury Surveillance (IRIS) project. The IRIS project is a comprehensive injury surveillance system and currently monitors the injury epidemiology of 1125 amateur adult rugby players from 25 male and female clubs in Ireland [24].

Differences between professional and amateur rugby in terms of injury epidemiology, injury incidence rates, physical characteristics of players, and challenges to contend with have been highlighted in previous research [7]. The majority of the world's rugby playing population, however, compete at an amateur level. Despite this, there is a paucity of information in relation to monitoring training at this level and the inherent challenges this presents [7]. There is a need, therefore, to examine the amateur game as its own distinct entity where staff and resources are reduced [4,7]. The current study extends the work of the existing IRIS project with the purpose of (1) developing an online TMS for amateur rugby using information from the existing literature and the current practices of S&C coaches, (2) evaluating the effectiveness of the system with the aim of recommending future methods of monitoring training, and (3) highlighting the challenges that can impact the effectiveness of a TMS that end-users (i.e., coaches and players) should take into consideration. The information gathered may be used as a guide for practitioners in helping develop successful approaches to monitoring training with amateur rugby teams. Additionally, the data regarding effective practices, as well as challenges met, may inform the development of monitoring systems in future investigations.

## 2. Methods and Materials

The experimental approach to the problem consisted of five phases: (1) establishing the current training and TL monitoring practices of amateur rugby union teams, (2) designing and developing the TMS, (3) recruiting teams and subsequently introducing the TMS, (4) supporting the S&C coaches using the TMS, and (5) evaluating the TMS. The study was conducted in accordance with the Declaration of Helsinki. All subjects provided informed consent before they participated in the study. Ethical approval for this research was provided by the University of Limerick's Institutional Research Ethics Review Board (project approval code: 2016_06_19_EHS, date of approval: 21 April 2020).

### 2.1. Current Training Monitoring Practices

In Ireland, there are 50 male and 7 female senior clubs at the national top tier of amateur adult rugby. To establish the current TL monitoring practices of these clubs, an online survey was designed and disseminated to 33 (n = 31 male and n = 2 female) S&C coaches representing 62% of the total number of male clubs and 71% of female clubs. Details of the design and results of the "Training load monitoring in amateur Rugby Union: A survey of current practices" are described by Griffin et al. [7].

### 2.2. The Development of Training Monitoring System

The measures chosen for inclusion in the TMS were established based on the results of Griffin et al. [7] and an extensive review of the literature. Subjective measures (e.g., visual analogue scales, perceived wellness/stress questionnaires) are more heavily relied upon in the amateur setting,

likely due to ease of use, accessibility, and low cost [1,4,7]. Additionally, self-reported player ratings of wellness provide a useful tool for coaches and practitioners to monitor player responses to the demands of training, competition, and life as an athlete [25]. This led to the decision to develop the TMS system using subjective measures exclusively. As the TL system was online, players could access the system through their smartphone device. Each team's S&C coach was also given an electronic tablet computer (iPad Pro, Apple Inc., Cupertino, CA, USA) to help with player data input. In order to log into the TL system, the players had to firstly enter their team's unique 4-digit pin followed by the player's own individual 4-digit pin. The additions made to the IRISweb system are outlined in Figure 1.

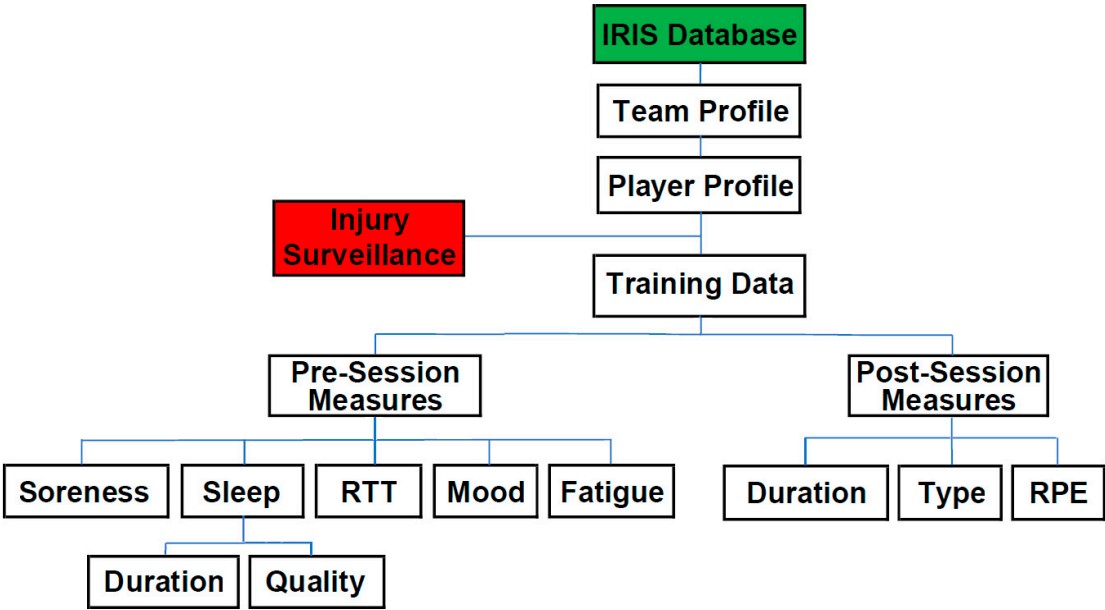

**Figure 1.** Structure of training monitoring system. IRIS = Irish Rugby Injury Surveillance project. RTT = Readiness to train. RPE = Rate of perceived exertion.

### 2.2.1. Pre-Session Measures

Players recorded six subjective measures prior to each training session and match. These consisted of fatigue, muscle soreness, sleep duration, sleep quality, mood, and readiness to train (RTT). A chromatic coloured 1–5 Likert scale was used to calculate each measure, with green indicating the most positive outcome and red indicating the most negative. A five-point Likert scale has been used to assess subjective measures of well-being in several previous articles [26–29]. The system used an automatic timestamp each time the player logged into the system to record their pre-session measures. A screenshot of the pre-session measures of the TMS can be seen in Figure 2.

Fatigue is difficult to define, but it can be characterised as an overwhelming sense of tiredness and lack of energy resulting from mental or physical exertion or illness [30,31]. In sport, it is often manifested in the difficulty in performing voluntary tasks or failure to maintain the required force or power output [30,31]. Accumulation of fatigue can result in a decrease in performance and an increase in injury risk and therefore monitoring fatigue is essential to understand the impact of TL [32].

Muscle soreness is often described as the soreness perceived by the athlete, presented with muscle stiffness and/or muscular tenderness [33]. The findings of McLean et al. [34] suggest that inappropriate TL can negatively impact on perceived muscle soreness resulting in a decrease in performance. It is therefore justifiable that the measure is commonly utilised as an important indicator of an athlete's recovery state [35].

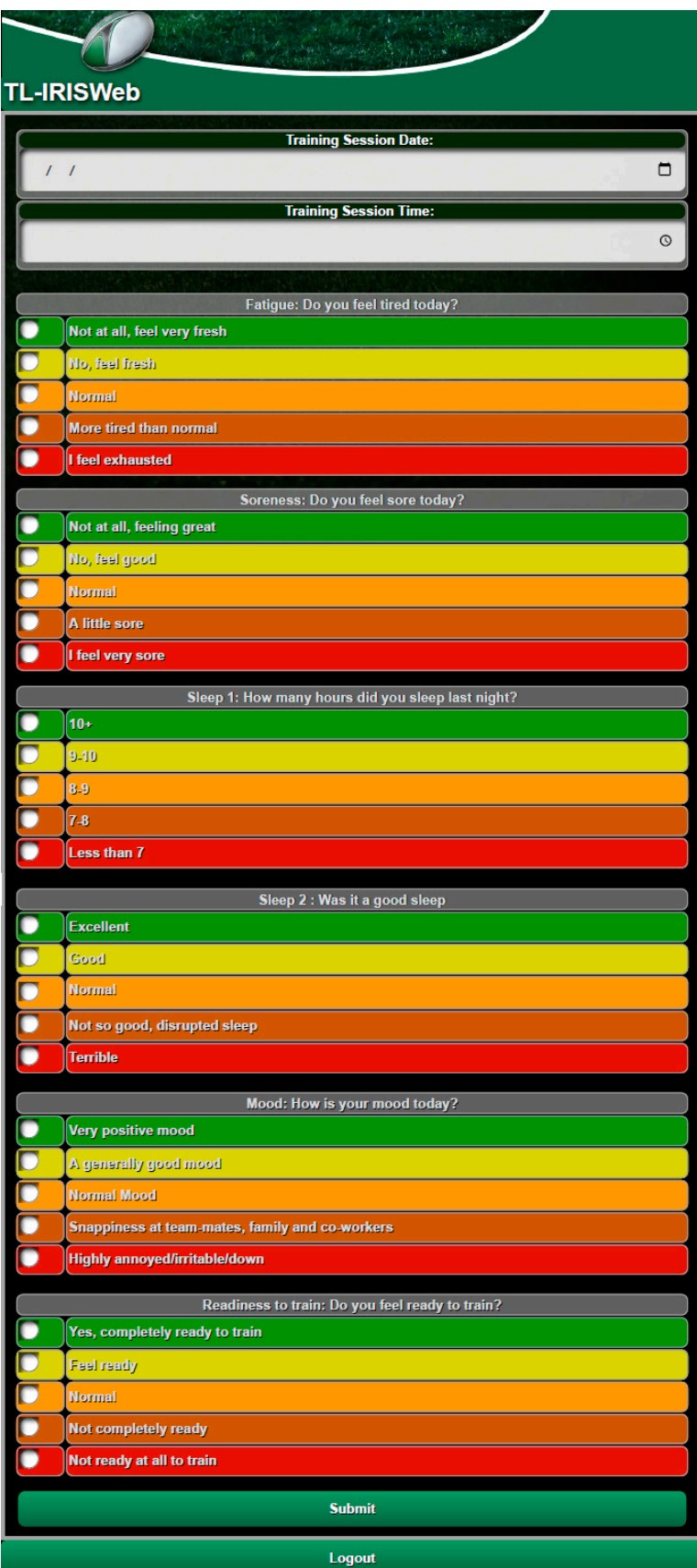

**Figure 2.** Screenshot of the pre-session page of the training monitoring system.

Poor sleep quality and short sleep duration can have a significant negative influence on sports performance and risk of injury and illness [26,36,37]. For instance, consistently obtaining less than 8 h of sleep has been shown to have 1.7 times greater risk for injury compared to those who slept for 8 h or more [38]. It has been suggested that practitioners working with athletes should monitor sleep quality and duration to improve overall health and performance [39].

Mood can be described as a simultaneous collection of affective states that subtly influence our experience, cognition, and behaviour [40]. Mood disturbance is often an early marker of unexpected and inappropriate TL and is therefore a useful inclusion in athlete monitoring practices [1]. Mood has also been measured on a five-point scale in previous research [34].

RTT is the relative preparedness of an athlete to accept a load [5]. RTT is commonly measured by means of a questionnaire and has been shown to detect the negative consequences of training [41]. No previous study, to the best of the authors' knowledge, used a Likert scale to measure RTT.

### 2.2.2. Post-Session Measures

Within 12 h of training or match completion, players recorded the type of session it was, the duration of the session, and the rating of perceived exertion (RPE) measured on a chromatic coloured 0–10 scale [42]. The options for session type included gym, fitness only, skills only, mixed session (fitness and skills), rehab/prehab, and match. Session duration was measured on a scale from 10–180 min in increments of 5 min. RPE is both a very common and valid method of measuring internal training intensity [7,43]. When session duration in minutes is multiplied by the RPE the result is known as the session rate of perceived exertion (sRPE) [44]. This method of calculating TL has been shown to be valid and reliable across a wide variety of exercise modalities [44]. A screenshot of the post-session measures of the TMS can be seen in Figure 3.

### 2.3. Recruitment and Introduction to the Training Monitoring System

The "Training load monitoring in amateur Rugby Union: A survey of current practices" survey included a question that gauged S&C coaches' interest in participating in a future study aiming to examine the relationship between training load and injury [7]. The rugby clubs that firstly expressed interest and secondly were using the IRIS injury surveillance system were the intended cohort to initially implement an online TMS. The S&C coaches were contacted electronically with an introduction to the study, the study aims, method of data collection, and data confidentiality. Players from seven men's senior teams were recruited to use the TMS for the 2019–2020 rugby season.

The TMS was introduced to the players and coaches by means of a 20-min presentation, given in person by the primary author or via video recording. This presentation outlined the purpose and potential benefits of their participation, the definition and explanation of each of the measures included, and a demonstration of how to use the system.

### 2.4. Weekly Training Report

In order to incentivise the use of the system and support the S&C coaches, a weekly training report was emailed to each team S&C coach at the end of the week. The report included a summary of each player's individual data and selected team mean data that included an outline of training type, the pre-session subjective measures, and acute:chronic workload ratio (rolling average model and exponentially weighted moving average model). The structure of the report was decided upon based on conversation with the S&C coaches prior to the commencement of data collection. The coaches were not provided with any commentary on the results.

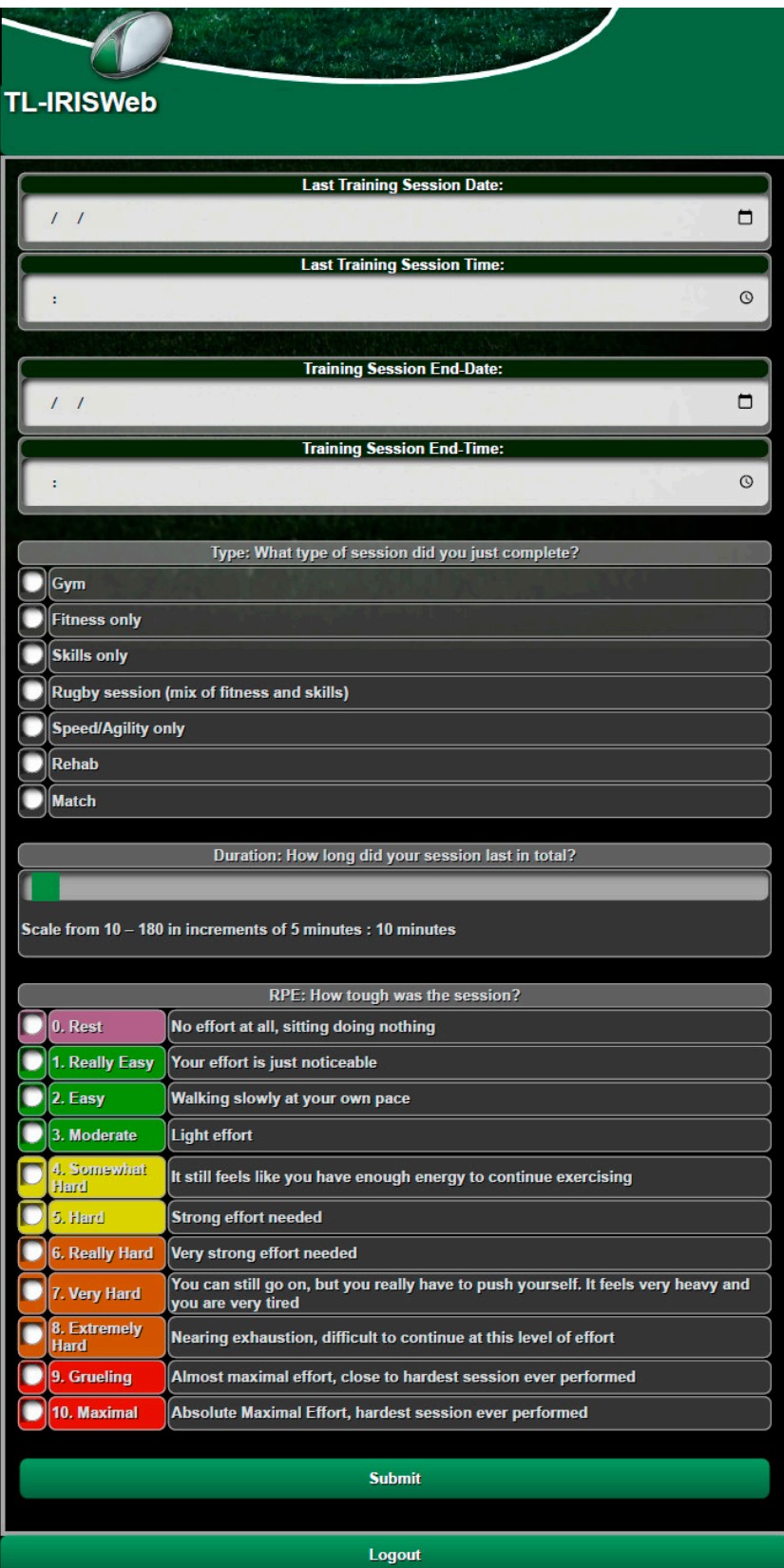

**Figure 3.** Screenshot of the post-session page of the training monitoring system.

### 2.5. Evaluation of the Training Monitoring System

Qualitative methods were used in order to evaluate the effectiveness, functionality, and application of the TMS, following a season of data collection. This consisted of (1) (Appendix A) a semi-structured interview and survey with the S&C coaches that used the system and (2) (Appendix B) a cross-sectional survey designed to evaluate the players' experiences of using the TMS.

In total, 72 players and 6 S&C coaches (age = 30 ± 5 years, coaching experience = 5 ± 2 years) from 6 senior clubs at the national top tier of amateur adult rugby participated in the study. One club that had initially used the TMS opted not to participate in its evaluation. All respondents were informed of the benefits and risks of participating, and written informed consent was obtained before undertaking the survey and interview in compliance with the Declaration of Helsinki.

At the end of data collection from the 2019–2020 season, the S&C coaches of all the teams using the TMS were contacted electronically with an invitation to participate in a semi-structured interview and survey. Interviews were conducted with the primary author by video call or phone call due to COVID-19 pandemic restrictions. The S&C coaches were also requested to share a weblink to the player survey with the players via the team messaging platform. Additionally, the email explained the purpose of the interview/survey, the time commitment, confidentiality of all collected information, and the respondent's entitlement to exit at any time without implication. The semi-structured interview guide and the survey were initially pilot tested with an advisory group of four academics experienced in this field of work, and each were asked to provide feedback on (1) whether or not the survey matched the purpose of the study, (2) the content of the questions, and (3) the structure of the survey. Several revisions were made to improve the clarity and specificity of the survey (e.g., addition of questions, rephrasing of questions). The survey responses were exported to Microsoft Excel (Microsoft, Redmond, WA, USA) for analysis.

#### 2.5.1. Coach Interview

The semi-structured interview guide consisted of six sections: (1) explanation of the purpose and rationale of the study, (2) coach and team profile, (3) coach and player system use, (4) system uptake and effectiveness, (5) coach's use of the data, and (6) challenges met and changes recommended going forward. It included a range of open-ended questions regarding factors influencing the implementation of the TMS (Table 1). The interview also included 10 closed questions. All interviews were audio recorded and then transcribed. Thematic analysis was employed to the qualitative data following the guidelines set out by Braun and Clarke [45] including: (1) familiarisation with the data, (2) generating initial codes, (3) searching for themes, (4) reviewing themes, (5) defining and naming themes, and (6) producing the report.

#### 2.5.2. Player Survey

The players' survey, "IRISweb Training Monitoring System Player Evaluation Survey", consisted of 10 closed questions and 3 open questions and was self-administered online via Qualtrics Survey Software (Qualtrics Labs Inc., Provo, UT, USA). Questions were aimed at gathering data regarding the overall experience, effectiveness, functionality, burden, and recommended changes to the TMS. Reponses were analysed using frequency analysis for each question and presented as absolute frequencies and percentages. Incomplete responses were excluded from all analyses. Mean response (±SD) was calculated for questions 1–6 and 8 as they used a 1–5 Likert scale. Questions 9 and 10 required respondents to rank chosen factors in order of importance and a scoring system identical to that of Griffin et al. [7] was used, whereby the top ranked answer received 7 points and the 7th ranked answer received 1 point (Table 5). Thematic analysis was also employed to the qualitative data gathered from the open-ended questions of 7, 12, and 13 following the guidelines set out by Braun and Clarke [45].

**Table 1.** Structure of semi-structured interview guide and examples of prompts used in interviews with strength and conditioning coaches.

| Section | Example Prompts |
|---|---|
| Coach and team profile | *Was this your first season with the team?*<br>*Were there any training monitoring practices in place prior to this season?*<br>*What was the team's typical weekly training schedule?* |
| Coach and player system use | *What were your methods of introducing the system to the team?*<br>*How did you attempt to ensure compliance from players?*<br>*What were you hoping to get out of using the system?* |
| System uptake and effectiveness | *How effective was the training monitoring system in terms of (1) enhancing player performance (2) reducing injury risk and (3) training load prescription and training design?*<br>*How useful did you find the pre-session measures in terms of (1), (2) and (3) above?*<br>*How useful were the post-session measures in terms of (1), (2) and (3) above?* |
| Coach's use of the data | *Did the data influence your training practice and prescription?*<br>*How was information fed back to the players?*<br>*Did you have the support from other coaching staff?* |
| Challenges met and changes recommended | *What challenges or problems did you meet using the system?*<br>*What do you think was the extent of burden placed on you and the players?*<br>*What changes to the system do you think are needed?*<br>*Would you use the system again next season?* |

## 3. Results

### 3.1. Evaluation of the Training Monitoring System

#### 3.1.1. Coach Interview

Six S&C coaches completed the semi-structured interview and survey, resulting in a response rate of 86%. All teams followed a similar typical weekly training schedule, whereby they completed a collective resistance training session on Mondays, a collective field-training session on Tuesdays and Thursdays, a match on a Saturday, and individual resistance training interspersed during the week. When asked if they had any TMS in place in previous seasons, only coach 6 replied yes, stating that they had *"a leaders group and they used to give me RPEs after every session which I would use for my tracking and monitoring across the team"*.

The thematic analysis of the S&C coaches' opinions of the effectiveness of the TMS resulted in six distinct higher order themes outlined in Table 2. The four most prominent themes were: training prescription and design, injury risk reduction, usefulness of pre-session measures, and usefulness of sRPE (all with four coach responses).

Six distinct higher order themes were identified from the thematic analysis of the S&C coaches' description of what challenges and issues they met while using the TMS (Table 3). The three most prominent themes were: Lack of player compliance, data inconsistency, and match-day challenges (all with six coach responses).

In terms of introducing the system and getting compliance from the players, all six S&C coaches stated that they found the presentation successfully outlined the purpose and potential benefits of the TMS and demonstrated how to use it. All six S&C coaches stated they thought most players understood the benefits of using the system and did not think how the system was introduced had a major impact on the lack of compliance and consistency of the players. When asked during the interview if they would use the system again if it was available in future seasons, two coaches stated that they would use it again, two coaches stated they would use it again if they felt that they could get compliance from all the players, and two coaches stated that they likely would not use it again due to the work involved in getting players to use the system.

**Table 2.** Higher order themes identified from strength and conditioning coaches' responses to questions regarding the effectiveness of the training monitoring system.

| Theme | Representative Quotes | No. of Responses |
|---|---|---|
| Training prescription and design | *"You can have GPS systems and heart rate and those things, but I find the subjective measures really help you know when you can push on or when you need to take a step back in terms of load. There were a few occasions where I saw things in the data that resulted in me talking to the coaches and the player and adjusting training."* (Coach 2)<br>*"We tweaked our training a little bit based on the data, particularly if we noticed a hard week."* (Coach 3)<br>*"It made me look at chronic load more and ACWR more than what I had previously done."* (Coach 6) | 4 |
| Injury risk reduction | *"The data definitely started some conversations that could have potentially prevented injury. For instance, the sleep data was useful to see that a lot of other players weren't getting enough sleep, so we were able to address that. Again, it started a conversation with those players. So, it did help with changing some behaviours that may have helped prevent injuries."* (Coach 1)<br>*"Being able to know if a session was an 8 or 9 RPE and then knowing to pull back for the next session will help with injury risk. It can also help to ensure that the players aren't overloaded coming into a big match and that there is enough recovery before it. I think it can really help at reducing injury when you have compliance and when the data is used in the right way."* (Coach 2)<br>*"It was really good and likely prevented injuries."* (Coach 3) | 4 |
| Enhancing player performance | *"What it was most useful for was after the tough Tuesday night session, if players were being flagged we could pull them out of that session and have them fully recovered and ready for selection on Saturday … maybe give them extra recovery work instead of training and see if we can get them right for the match. In that regard it can be very useful for match preparation. It is better we know early so we can be prepared rather than him pulling up in the warmup or early in the match."* (Coach 4)<br>*"It can also help to ensure that the players aren't overloaded coming into a big match and that there is enough recovery before it."* (Coach 2) | 3 |
| Usefulness of pre-session measures | *"I found the pre-session measures useful because the players that are working all day are coming to training and telling you how prepared they are for training and by giving me numbers for each measure I can then act on it by cutting them from certain parts of the training or whatever I think is necessary. That is massively beneficial. It's a great conversation starter."* (Coach 6)<br>*"I thought the 1 to 5 scale worked quite well and I especially liked that it was coloured. When I was checking the data before the session the reds and oranges jumped out at me. When I saw them I knew I had to address it straight away before the session started. But the fact that it was coloured gave a good gauge of where the team was at. You might only have 10 min before the training session to look at the data, so the colours really help."* (Coach 1)<br>*"They tell me how up for the session they are, and its score would capture the way I would then communicate to them."* (Coach 5)<br>*"They were really effective as a conversation starter."* (Coach 1) | 4 |
| Usefulness of sRPE | *"The sRPE data is the most useful for training design and prescription. It allows me to look back at the previous data and plan the future sessions."* (Coach 5)<br>*"Very important measure for us because it's cheap, time efficient, and it gives you a solid number for load at the end of it."* (Coach 6) | 4 |
| Potential of weekly report | *"If the players had been more consistent it would have been really useful."* (Coach 1)<br>*"If we had better buy-in across the panel the training load report would have been really useful."* (Coach 3) | 3 |

**Table 3.** Higher order themes identified from strength and conditioning coaches' responses to questions regarding challenges and issues met using the training monitoring system.

| Theme | Representative Quotes | No. of Responses |
|---|---|---|
| Lack of player compliance | *"Even after the presentation they didn't all see the value in it even though it was well explained to them. I know that was a problem and I heard players mentioned that."* (Coach 1) <br> *"I'm not sure the players understood that the reason we were collecting the data was to help them, it wasn't to keep ourselves busy. I suppose it's about changing their belief system and teaching them that monitoring is important for injury prevention and increasing their performance."* (Coach 2) <br> *"I think maybe the unknown was probably an issue. Some of the players just weren't used to it, no fault on their part, they have just never been exposed to it … Then after a while my encouragement to get them to use it stopped because it was becoming too much work."* (Coach 4) | 6 |
| Data inconsistency | *For the first couple of weeks there was good buy-in but then a lot of the players that were using it dropped off when they saw the other players not using it … I suppose us not acting on the data caused players to drop out but again that was because we weren't getting consistent data. We weren't able to get a clear picture of the whole squad so we didn't feedback the data to the players as much as we should have and take action on the data."* (Coach 1) <br> *"Getting them to do it consistently is key even if it means you don't get all the measures you want. If it becomes too much of a chore they won't do it."* (Coach 2) <br> *"Because we had such a small group of players who were consistently using the system the data kind of got skewed a little bit. If we had better buy-in across the panel the weekly training report would have been really useful."* (Coach 3) | 6 |
| Match-day challenges | *"Getting the players to give data on match days was a disaster. There are so many psychosocial factors going on. I didn't like asking them to think about their state pre-game because it might highlight something negative in them … I just felt it was impractical as I wasn't even going to be doing anything with the data at that stage."* (Coach 1) <br> *"Getting players to fill out their sRPE after a match is a nightmare. It's a challenge because that is going to be the most difficult session of the week and probably the most important data to get."* (Coach 3) <br> *"It's near impossible. It is very difficult to approach them before a game and I wouldn't dare approach them asking them to fill in their data after a game if we lost. They don't want to talk about it."* (Coach 6) | 6 |
| External Confounders | *"In the amateur game, you can give them all the guidelines you want but because there are so many other factors, work, home life etc. that you can't be guaranteed they'll follow them … There are too many variables in the amateur game to control for. I think there's a tendency to think that what works in the pro game transfers to the amateur game and that we should try to do exactly what they are doing but a lot of the time it doesn't work."* (Coach 1) <br> *"I think it didn't matter what we did this year just wasn't going to work. This year nothing was right; eating, drinking, their behaviour off the field, nothing was right compared to the previous years. You could have said I'll pay you a grand each to use the system and I don't know if it would have made any difference."* (Coach 2) <br> *"We've quite a few guys who are in college but have part-time jobs at night, a lot of them in pubs. On Wednesday nights they could be working until 3 in the morning and then getting up for a lecture at 9. Then we were wondering why they were training so poorly on a Thursday night."* (Coach 3) <br> *"Because so many of our players are working class, doing jobs like labouring or construction, really taxing work on the body, they were the poorest at filling in their TL data but they are probably the ones you want the most. They'd have no problem having a chat with you about how they were feeling physically but to get them to actively fill in something is a struggle."* (Coach 6) | 6 |
| Lack of support | *"The other coaches did buy-in in the sense they knew why it would be good to use but at the same time they left it to me. They didn't want to worry too much about it."* (Coach 4) <br> *"The head coach was very supportive of it, but the skills coach was not supportive of it whatsoever. That just wasn't good enough because everything starts at the top. Fortunately, the head coach really believed in it … In my opinion for a monitoring system like this to be effective and get total adherence every coach needs to be 100% on board."* (Coach 6) <br> *"Trying to chase players to fill in data and then trying to collate and analyse the data when you just don't have the time to do it and more importantly you're not getting paid to put in that extra time."* (Coach 6) | 3 |
| Player log-in issues | *"The log-in tended to be an issue for the players, so not so much the system itself but the fact that they had to have a team PIN and their own PIN. That tended to cause issues for players, forgetting their passwords and having to be given it numerous times"* (Coach 1) <br> *"Once you were in the system it was phenomenal, five out of five, really easy to use and really useful data but the getting in with the PINs was a barrier."* (Coach 3) | 3 |

As part of the survey, the S&C coaches were asked to rate seven aspects of their experience using the TMS on a 1–5 Likert scale (1 = the most negative response, 5 = the most positive response). The mean responses (±SD) are represented in Table 4.

**Table 4.** Strength and conditioning coaches' mean responses (±SD) to questions using a Likert scale from 1–5 (1 = the most negative response, 5 = the most positive response) relating to the training monitoring system.

| Ease of Use | Interface | Enhance Training Prescription | Reduce Injury Risk | Enhance Performance | Regularity of Feedback | Regularity of Data Use |
|---|---|---|---|---|---|---|
| 3.2 ± 1.0 | 4.2 ± 0.8 | 3.5 ± 0.8 | 3.5 ± 1.0 | 3.3 ± 0.5 | 2.0 ± 0.6 | 2.0 ± 0.6 |

The S&C coaches ranked the seven measures used in the TMS in order of importance to their training prescription practices and reduction of injury risk throughout the season (Table 5). Five coaches ranked sRPE as the most important in terms of training prescription practices (40 arbitrary units (AU)) while sleep duration (28 AU) and muscle soreness ranked (27 AU) second and third, respectively. Similarly, four coaches ranked sRPE as the most important in terms of reduction of injury risk (39 AU), again followed by muscle soreness (30 AU) and sleep duration (27 AU). All six S&C coaches stated they would keep sRPE when asked what three measures they would keep if the TMS had to be streamlined, followed by muscle soreness (5 coaches), sleep duration (4 coaches), RTT (2 coaches), fatigue (1 coach), and no coach opted to keep sleep quality or mood.

**Table 5.** Ranking of the pre-session and post-session measures by the players (*n* = 72) and strength and conditioning coaches (*n* = 6) *.

| Measures | Coaches | | | | Players | | | |
|---|---|---|---|---|---|---|---|---|
| | Training Prescription | | Injury Risk | | Preparation | | Injury Risk | |
| | Rank | Score (AU) | Rank | Score (AU) | Rank | Score (AU) | Rank | Score (AU) |
| Muscle Soreness | 3 | 27 | 2 | 30 | 1 | 359 | 1 | 383 |
| RTT | 5 | 21 | 5 | 22 | 3 | 300 | 4 | 309 |
| Sleep Duration | 2 | 28 | 3 | 27 | 7 | 223 | 6 | 203 |
| Sleep Quality | 4 | 25 | 4 | 25 | 6 | 244 | 5 | 252 |
| Mood | 7 | 9 | 7 | 8 | 5 | 253 | 7 | 167 |
| Fatigue | 6 | 18 | 6 | 17 | 2 | 350 | 2 | 352 |
| sRPE | 1 | 40 | 1 | 39 | 4 | 287 | 3 | 350 |

* AU = Arbitrary Units, RTT = Readiness to train, sRPE = Session rating of perceived exertion. Note: Measures are ranked from 1 (most important) to 7 (least important). Maximum possible score for measures: player survey = 504 AU, coach survey = 42 AU.

### 3.1.2. Player Survey

Players were asked to rate seven aspects relating to their experience of using the TMS using a Likert scale from 1–5 (1 = the most negative response, 5 = the most positive response) and the frequency analysis is represented in Figure 4. The methods of feedback given included one-to-one conversations (43%), text message (35%), and collectively at training sessions/meetings (22%). Players were asked how often they or their coach changed their usual training/match preparation based on their data and the mean score was 2.3 ± 1.1, with 1% stating that it always occurred, 21% most of the time, 10% about half of the time, 47% sometimes, and 21% stating that it never occurred.

Players ranked the seven measures used in the TMS in order of usefulness to their own training/match preparation, with muscle soreness receiving the most first preference selections (24%) and ranking first with a total score of 359 AU out of a maximum score of 504 AU (Table 5). Fatigue ranked second (350 AU) followed by RTT (300 AU), sRPE (287 AU), mood (253 AU), sleep quality (244), and lastly sleep duration (233 AU). When asked to rank the seven measures in order of usefulness to reducing their risk of injury, again muscle soreness received the most first preference selections (28%) and ranked first with a total score of 383 AU (Table 3). Fatigue ranked second (352 AU) followed by sRPE (350 AU), RTT (309 AU), sleep quality (252), sleep duration (203 AU), and lastly mood

(167 AU). Players were asked if the system was streamlined, what three measures would they keep and muscle soreness was selected by 75% of players, followed by sRPE (65%), fatigue (47%), RTT (44%), sleep quality (26%) mood (26%), and sleep duration (15%).

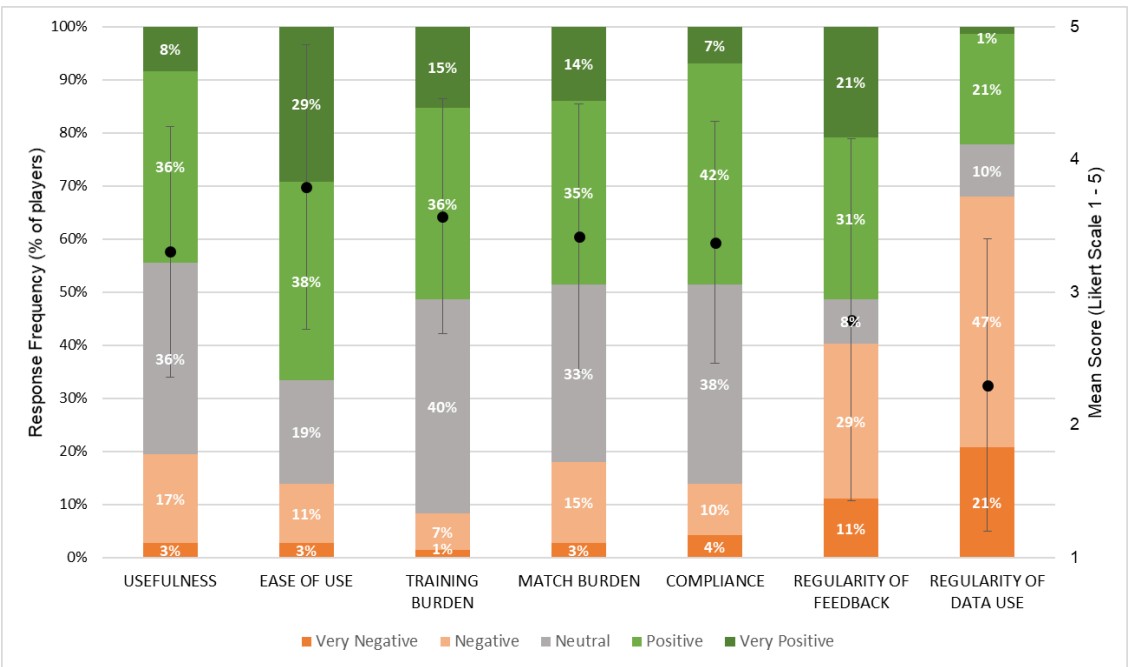

**Figure 4.** Players' responses to the 1–5 Likert scale questions (1 = the most negative response, 5 = the most positive response) evaluating the training monitoring system.

When asked if they encountered any problems or difficulties using the system during the season, 72% of players stated that they encountered none, 24% stated they had difficulties with the process of logging into the system, and 4% stated they had difficulty remembering to use the system. Twenty-one percent of the surveyed players responded to the open-ended question regarding changes required to the system and two higher order themes were highlighted: (a) feedback and use of data (b) reminders. Fourteen percent of players stated they wanted to receive more detailed and frequent feedback on the data they had given and to see that data contribute to training prescription. One player reported: *"I would like more feedback on the data collected and this to be discussed very quickly as part of our training"* while another stated he *"didn't really know if the monitoring system was looked at by rugby coaches"*. Seven percent of players stated they would like to see the addition of regular reminders to the system as highlighted by the player's quote: *"give reminders to complete before and after training"*.

## 4. Discussion

This study outlines the design and justification, development, implementation, and subsequent evaluation of an online TMS to be used by amateur rugby union teams. The S&C coaches highly valued the measures of sRPE, muscle soreness, and sleep duration while the players found muscle soreness, fatigue, sRPE, and RTT most useful. Conversely, the main barriers to effective training monitoring from all six coaches' perspective are a lack of player compliance, data inconsistency, gathering data around matches, and external confounders. The main barriers from the players' perspective are a lack of feedback and evidence of data use from their coaches. However, 73% of S&C coaches working with teams playing at the highest level of amateur rugby union in Ireland monitor training [7], and this current study offers both a method that can be effective, and pitfalls to avoid for both practitioners and researchers alike.

### 4.1. Effectiveness of the Training Monitoring System

It is evident from the coach interviews that the lack of consistent data negatively impacted the effectiveness of the TMS. Despite this, the mean S&C coaches' responses to its usefulness in terms of reducing injury risk was 3.5/5 and 3.3/5 for enhancing player performance. This is almost identical to the results of the survey of Griffin et al. [7]. As five of the six coaches did not have a TMS in place prior to this season, the system outlined in this study offers a comparatively effective method to that currently being used by other amateur rugby teams. The effectiveness of this system could be improved upon, however, if the issues described in Table 3 are addressed (e.g., match day challenges, lack of support). The four S&C coaches with the greatest adherence from players spoke very positively about the system in terms of its ability to inform training prescription and design and help reduce injury risk (Table 2). Additionally, three coaches referred to its usefulness in enhancing performance by aiding players' match preparations.

### 4.2. Subjective Measures of Monitoring Training

As described in the methods section, justification for the measures included in the TMS was based largely on previously published research [1,4,5,31,34,39]. This study gives further support to the use of subjective measures as a method of monitoring training [1,4]. Lack of consistency of data collection aside, four S&C coaches found value in the pre-session measures on a coloured five-point Likert scale as a method of starting conversations with the players and, importantly, knowing which players in particular to prioritise speaking with in order to get more context around their score. Due to the external confounders associated with amateur-level athletes (e.g., their occupation), having this information will inform the preparedness of the player for the upcoming training session or match. From the coaches' perspective, sRPE ranked highest in order of importance to both their training prescription practices and attempts to reduce injury risk throughout the season followed by muscle soreness and sleep duration (Table 5). This is consistent with the results of the question regarding the streamlining of the system as these three variables again received the greatest selection. Similarly, the players ranked muscle soreness most useful to both their own training/match preparation and to reducing their risk of injury. However, in contrast, fatigue ranked second and sRPE and RTT were ranked next. Coaches saw less value in the fatigue and RTT measures compared to the players. Neither players nor coaches ranked the mood or sleep quality measures highly. Although all seven measures are validated based on previous research [1,4,5,31,34,39], the results of this study suggest that muscle soreness, sRPE, fatigue, sleep duration, and RTT may be more valuable in a practical setting.

A monitoring system should be as efficient and succinct as possible to reduce player and data analysis burden and in turn increase the consistency of the data gathered [6]. Consequently, only measures that will inform the coach's training practices regularly should be included in the system. Its first-place ranking in relation to both training prescription and design and injury risk (Table 5) combined with the S&C coaches' representative quotes (Table 2) further demonstrates the usefulness of sRPE as an effective monitoring tool [4,14,44].

### 4.3. Lack of Universal Training Monitoring System

The varying nature of the S&C coaches' replies to the question of whether they would use the system again if available demonstrates that an online TMS can be beneficial to the practices of S&C coaches but not all will use it due to the associated challenges. Two coaches stated that they would use it again, two coaches stated that they would not use it again, while two coaches stated if they could get compliance from the players and therefore consistent data, they would use it again. The findings of this study have shown that a system of this nature can be effective and beneficial to many teams. However, it is clear that an open relationship with the players that facilitates honest conversations regarding their preparedness and TL capabilities should be the foundation of any player monitoring system [46]. Practitioners should still see this as a priority when developing their TMS. An online

TMS can provide information that triggers these conversations, which in turn has the potential to support player performance and mitigate injury risk [46]. This is evident from the representative S&C coaches' quotes in Table 2, for example, *"That is massively beneficial. It's a great conversation starter."* (coach 6). Research should guide training monitoring strategies; however, conditions may prevent its implementation resulting in the necessity of adjusted methods [47]. Coach 1's quote, *"I think there's a tendency to think that what works in the pro game transfers to the amateur game and that we should try to do exactly what they are doing but a lot of the time it doesn't work"* is an example of the many quotes from Table 2 that supports this conclusion.

### 4.4. Challenges Associated with Monitoring Training

The player survey results suggest that the majority of players found the online TMS useful, with 80% of respondents rating it as either moderately, very, or extremely useful (Figure 4). However, it is evident from Table 3 that the S&C coaches found several challenges with using a TMS of this nature.

### 4.5. The Player Compliance—Coach Feedback Loop

The greatest challenge for S&C coaches may be getting initial and then continued compliance from players to ensure consistent data. All six coaches found the introductory presentation successfully demonstrated the purpose, potential benefits, and functionality of the system. It appears that, for it to be effective, players need to see value in using the monitoring system to the extent that it offsets the burden of using it. Two coaches suggested that lack of compliance was due to the team having a history of relatively low injury rates and players not being regularly required to play or train in a fatigued state, and therefore may not see the worth in using a system. However, Neupert et al. [46] found that the main reason for poor compliance from players was lack of feedback on their data and training modifications founded on said data. A limiting factor highlighted by all six coaches was the lack of consistency in the training data gathered, which resulted in difficulty supplying feedback to the players and adjusting the training prescription and design based on the data. Players rated the regularity of both receiving feedback on their data and the coaches using their data poorly, scoring 2.8/5 and 2.3/5, respectively. In fact, 10 players specifically highlighted the lack of feedback and lack of data use being a considerable deterrent to them using the system, with one player reporting he *"would like more feedback on the data collected and this to be discussed very quickly as part of our training"* while another stated he *"didn't really know if the monitoring system was looked at by rugby coaches"*. Coaches also rated the regularity of their feedback (2.0 ± 0.6) and data use (2.0 ± 0.6) quite poorly. This causes a quagmire, whereby in order for S&C coaches to act on data it must be consistent, but players must receive regular feedback and perceive that their data is being acted upon in order to give it consistently. The success of a TMS is largely based on its ability and opportunity to lead to the implementation of change in practices and the delivery of feedback to players and other coaches is critical to achieving this [6]. To combat this disconnect, S&C coaches should consider outlining the issue when introducing their TMS and then scheduling regular collective feedback sessions throughout the season that show evidence of how the data is being used. The consensus statement on monitoring athlete TL by Bourdon et al. [48] states that, for a TMS to be successful, a clear feedback loop that provides educational information to athletes needs to be established along with regular intervention if special circumstances arise. Additionally, an online system such as that used in the current study may be improved upon if the data analysis was real-time instantaneous within the software to allow S&C coaches to make in-time decisions on the data and feedback to players. Due to the challenge of achieving consistent data, future research should examine methods of addressing missing TL data and possible data imputation methods.

### 4.6. Support to Strength and Conditioning Coaches

Another challenge that restricts the TL monitoring practices of S&C coaches working with amateur rugby teams is the lack of resources, time, and financial compensation [7]. The findings of this study further emphasise this limiting factor and it appears that, even when given a system and weekly report

this is not necessarily resolved. To mitigate this challenge, it is important that S&C coaches have the full support and assistance of all members of the coaching staff and club stakeholders to ensure optimal player compliance and to share the burden of monitoring TL [49].

### 4.7. Training Monitoring System Functionality

The system interface was rated highly by the S&C coaches (4.2 ± 0.8) and the ease of use was rated highly by the players (3.8 ± 1.1). However, from a functionality perspective, an issue highlighted by half of the coaches and 24% of the players was the log-in process using a 4-digit team and player PIN. It would appear that this created an additional barrier in some players uploading their data, with several suggesting a smartphone application being a more appropriate forum for the system. Training load monitoring using a smartphone application has been concluded to be a feasible and practical tool to be used outside of a controlled laboratory setting [50]. Furthermore, five players suggested the addition of automated reminders before and after training would be beneficial, which would again lend support to using a smartphone application.

### 4.8. Match-Day Challenges

The six S&C coaches interviewed spoke about the challenges associated with gathering TL data around matches, resulting in the theme of match-day challenges being highlighted. The consensus of the coaches was that getting pre-session measures before a match was impractical as players have their own routines and coaches do not want to highlight negative thoughts within the players. In addition, the coaches do not see the benefit of having these data as they are not going to act on it before the match. There is, however, evidence to suggest that a pre-match measure of wellness can be used to indicate reactions to external load and hence assist player rotation strategies [51]. This may be particularly valuable in amateur rugby as a congested competition schedule can cause increased strain on players [52]. Coaches also stated they felt uncomfortable asking players to submit their post-match data, particularly after a loss. Similarly, 18% of players found giving data around matches to be very or extremely burdensome compared to only 8% around training sessions (Figure 4). This is similar to the findings of Akenhead and Nassis [53], who suggested low use of post-match sRPE may be due to the effect on sensitivity as a consequence of the psychological states of players and coaches in the competitive environment. Additionally, the match outcome (win/loss/draw) can have a moderate to large effect on the players' RPE [54]. Nonetheless, without post-match sRPE data, an S&C coach cannot accurately calculate weekly load and as a result training strategies and TL prescription based on these data will be limited [37]. Crucially, sRPE has been shown to be temporally robust for up to at least 24 h post-exercise [55] and a practical solution to alleviate the issue may be for practitioners to collect post-match sRPE the following day.

### 4.9. Confounders External to Sport

Issues outside of sport that impact on players' ability to train and perform were highlighted by all six coaches and led to the theme of external confounders being identified (Table 3). From the S&C coaches' perspective, these external cofounders (e.g., limited time with players, occupation of players, resources available, etc.) are more copious and predominant at the amateur level. This issue is epitomised by the quote of coach 1: "*In the amateur game, you can give them all the guidelines you want but because there are so many other factors, work, home life etc. that you can't be guaranteed they'll follow them ...* ". It is important to recognise that, despite the vast majority of rugby members playing at an amateur level, this is not reflected in the wider research in this area, with the majority being conducted in the professional setting [56]. It is evident from the findings in relation to the challenges maintaining player compliance highlighted by the S&C coaches in this study that the results and recommendations derived from professional sport are not always transferable to the amateur setting due to the various external cofounders. The International Olympic Committee consensus statement on load in sport and risk of injury suggests monitoring should be performed consistently to allow adjustment to TL prescription

but without overburdening the players [5]. This poses a particular challenge at the amateur level due to its external confounders and unique challenges. To aid training monitoring practices, future research is needed into the examination of the impact these external confounders have on injury and performance at the amateur level.

*4.10. Study Limitations*

A limitation of this study is that one of the clubs that used the TMS did not participate in the survey or interview. Since this club ceased use of the system mid-way through the season, the input and opinions of their players and coach would have added valuable insight into the assessment of the system. Additionally, qualitative interviews by means of phone call have generally been considered an inferior alternative to face-to-face interviews [57]; nevertheless, there is support for its use in obtaining rich data [57,58].

## 5. Conclusions

This study offers a method of monitoring training using an online system that is attainable, scientifically grounded, and both S&C coaches and players find moderately to very effective. Practically, pre-session measures can prompt conversations between coaches and players while sRPE can inform S&C coaches' training prescription and design. These measures combined can potentially promote a reduction in injury risk and ultimately support player performance. There are several challenges to monitoring training in the amateur setting that will limit its success and effectiveness, but potential resolutions to these have been outlined. A monitoring system should only include measures that the coach intends to act upon and feedback regularly to the players. S&C coaches should recognise that the main barriers impeding player compliance are the lack of feedback on their data and evidence of its use in training design and prescription. Importantly, in order for coaches to overcome these barriers, they need both initial and continued compliance from the players, resulting in sufficient data to act upon. This study gives practical guidance to practitioners and researchers aiming to monitor rugby training in an amateur setting.

*5.1. What Are the Key Findings?*

- Both the strength and conditioning (S&C) coaches and players alike value an online training monitoring system (TMS) but the greatest barriers to its effectiveness are lack of player compliance, data inconsistency, gathering match data, and external confounders outside of the sport.
- Muscle soreness, fatigue, sleep duration and readiness to train (RTT) were highly rated as effective pre-session measures and are particularly useful at signalling red flags that prompt conversations between coaches and players.
- Session rating of perceived exertion (sRPE) was a highly rated metric by S&C coaches and players for aiding training prescription and mitigating injury risk.

*5.2. What Are the Key Practical Applications for Coaches?*

- This study provides further support to the use of subjective measures as a method of monitoring training. Their ease of use, accessibility and low cost make them practically applicable in the amateur setting.
- For players to engage with a TMS and give consistent data, they need regular feedback and evidence that the data are informing their training. This should aid in achieving consistent data from players and in turn improve the effectiveness of the TMS.
- S&C coaches should be aware that the temporal robustness of sRPE may alleviate the difficulties around collecting post-match sRPE.

- It is important that S&C coaches have the full support and assistance of all members of the coaching staff to ensure player compliance and to share the associated training monitoring and analysis demands.

## 6. Data Availability

The datasets generated and/or analysed during the current study are available from the corresponding author on reasonable request.

**Author Contributions:** All authors contributed to the design and implementation of the research, to the analysis of the results and to the writing of the manuscript. All authors have read and agreed to the published version of the manuscript.

**Funding:** Funding for this study was provided by the Irish Research Council.

**Acknowledgments:** The results of the present study do not constitute endorsement of any product by the authors and the authors have no conflict of interest to disclose. The authors would like to acknowledge with considerable gratitude the members of the Irish Rugby Injury Surveillance (IRIS) Research team and the Irish Rugby Football Union (IRFU) for their help throughout the study period and the Irish Research Council for financially supporting this research.

**Conflicts of Interest:** The authors declare no conflict of interest.

## Appendix A. Copy of Coach Semi-Structured Interview and Survey

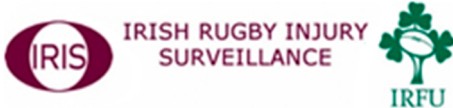

**Evaluation of the IRISweb training load system coach semi-structured interview and survey**

1. The interviewer explains the purpose and rationale of the study and describes the interview process. The participant is informed that participation is completely voluntary and that all responses are completely confidential and governed by our institutional ethics and GDPR.

2. **Background information**

- Pathway to their current position (previous experience, number of seasons with team, etc.)
- Typical weekly schedule with team
- Previous training load (TL) monitoring practices prior to this season

3. Coaches are asked to reflect on their experiences using the IRISweb training monitoring system. **Focus of the conversation will be on the following 4 key areas:**

- **Use of the system**

- ○ Methods of getting compliance from players (if good compliance was achieved)
- ○ Reasons for lack of compliance (if poor or no compliance was achieved)
- ○ Reasons for initial participation
- ○ Reasons for continual participation
- ○ Reasons for drop-out (if applicable)
- ➢ Survey questions 1 and 2 will be asked at this time

- **Effectiveness of the system**

- ○ The effectiveness of the TL monitoring in terms of

    **(1)** enhancing player performance
    **(2)** reducing injury risk

> **(3)**     TL prescription and training design

○ Usefulness of the weekly TL report delivered by the research team in terms of **(1)**, **(2)** and **(3)** above
○ Usefulness of the pre-session measures in terms of **(1)**, **(2)** and **(3)** above
○ Usefulness of the post-session measures in terms of **(1)**, **(2)** and **(3)** above
➢ Survey questions 3, 4, 5, 6 and 7 will be asked at this time

● **Use of the data**

○ How they used the data
○ Had the data an influence on their training practice and prescription
○ How did the data influence on training practice and prescription
○ How they fed back information to the players
○ What TL information did they feed back to the players
○ Support from other coaching staff (i.e., did the other coaches take on board the information given)
➢ Survey questions 8 and 9 will be asked at this time

● Changes required going forward

○ Challenges or problems they met using the system
○ Challenges their players met using the system
○ Difference in players' practices between match and training sessions
○ Extent of burden placed on them and the players
○ Changes needed to the system
○ Ideal TL monitoring practices in terms of **(1)**, **(2)** and **(3)**
➢ Survey questions 10 will be asked at this time

**Within the conversation, the following structured questions will be asked, and a quantitative response given:**

1.  Please rate the ease of use of the IRISweb Training Load Monitoring System.

| 1 | 2 | 3 | 4 | 5 |
|---|---|---|---|---|
| Very Poor | Poor | Neutral | Good | Very Good |

2.  Please rate the IRISweb Training Load Monitoring System interface.

| 1 | 2 | 3 | 4 | 5 |
|---|---|---|---|---|
| Very Poor | Poor | Neutral | Good | Very Good |

3.  Please rate how useful you found the IRISweb Training Load Monitoring System as a whole in terms of <u>enhancing player performance</u>.

| 1 | 2 | 3 | 4 | 5 |
|---|---|---|---|---|
| Very Poor | Poor | Neutral | Good | Very Good |

4.  Please rate how useful you found the IRISweb Training Load Monitoring System as a whole in terms of <u>reducing injury risk</u>.

| 1 | 2 | 3 | 4 | 5 |
|---|---|---|---|---|
| Very Poor | Poor | Neutral | Good | Very Good |

5. Please rate how useful you found the IRISweb Training Load Monitoring System as a whole in terms of TL prescription and training design.

| 1 | 2 | 3 | 4 | 5 |
|---|---|---|---|---|
| Very Poor | Poor | Neutral | Good | Very Good |

6. Please rank the following measures in order of importance to your training prescription practices throughout the season-1 being the most important to you and 7 the least important to you.

| Measure | Rating (1–7) |
|---|---|
| Muscle soreness | |
| Sleep duration | |
| Sleep quality | |
| Mood | |
| Fatigue | |
| Readiness | |
| Training Load (RPE x duration) | |

7. Please rank the following measures in order of importance to reduction of injury risk throughout the season-1 being the most important to you and 7 the least important to you.

| Measure | Rating (1–7) |
|---|---|
| Muscle soreness | |
| Sleep duration | |
| Sleep quality | |
| Mood | |
| Fatigue | |
| Readiness | |
| Training Load (RPE x duration) | |

8. Did you give feedback on the data gathered to the players?

| 1 | 2 | 3 | 4 | 5 |
|---|---|---|---|---|
| Never | Seldom | Somewhat | Often | Always |

9. Did you change your usual training/match preparation based on the data gathered?

| 1 | 2 | 3 | 4 | 5 |
|---|---|---|---|---|
| Never | Seldom | Somewhat | Often | Always |

10. If the IRISweb Training Load Monitoring System had to be streamlined, what 3 measures would you keep?

| Measure | Select 3 |
|---|---|
| Muscle soreness | |
| Sleep duration | |
| Sleep quality | |
| Mood | |
| Fatigue | |
| Readiness | |
| Training Load (RPE x duration) | |

**Appendix B. Copy of Player Survey**

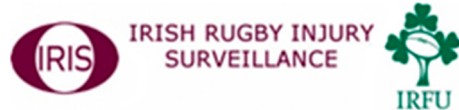

**Evaluation of the IRISweb training load system survey of players**

This short survey is being used to get your opinions on the IRISweb Training Load System so we can improve monitoring practices for domestic Rugby in Ireland. We are interested in your honest opinions. All responses are completely confidential and governed by our institutional ethics and GDPR. 13 questions with an estimated completion time of 4–6 min.

1.  Please rate how useful the IRISweb Training Load Monitoring System as a whole was to you and your team.

    ○   Extremely useful (1)
    ○   Very useful (2)
    ○   Moderately useful (3)
    ○   Not very useful (4)
    ○   Not at all useful (5)

2.  Please rate the ease of use of the IRISweb Training Load Monitoring System.

    ○   Very easy (1)
    ○   Easy (2)
    ○   Neutral (3)
    ○   Not easy (4)
    ○   Difficult (5)

3.  How burdensome/demanding did you find using the IRISweb Training Load Monitoring System before and after training sessions?

    ○   Not at all (1)
    ○   Not very (2)
    ○   Somewhat (3)
    ○   Very (4)
    ○   Extremely (5)

4.  How burdensome/demanding did you find using the IRISweb Training Load Monitoring System before and after matches?

    ○   Not at all (1)
    ○   Not very (2)
    ○   Somewhat (3)
    ○   Very (4)
    ○   Extremely (5)

5.  During the season, how compliant do you feel you were at using the IRISweb Training Load Monitoring System before and after all trainings and matches?

    ○   Never used it (1)
    ○   Rarely used it (2)

○ Used it sometimes (3)
○ Used it most of the time (4)
○ Always used it (5)

6. How often, if at all, did you receive feedback on your data from your Strength and Conditioning Coach?

○ Always (1)
○ Most of the time (2)
○ About half the time (3)
○ Sometimes (4)
○ Never (5)

7. Please supply detail of how you received this feedback (e.g., what feedback you received and how you received it).

______________________________________________________________

8. How often, if at all, did you or your coach change your usual training/match preparation based on your use of the IRISweb Training Load Monitoring System?

○ Always (1)
○ Most of the time (2)
○ About half the time (3)
○ Sometimes (4)
○ Never (5)

9. Please rank the following measures in order of usefulness to your own training/match preparation: 1 being the most useful to you and 7 the least useful to you. Note: drag and drop

______ Muscle Soreness (1)
______ Readiness to train (2)
______ Sleep Duration (3)
______ Sleep Quality (4)
______ Mood (5)
______ Fatigue (6)
______ Training Load (RPE x duration) (7)

10. Please rank the following measures in order of usefulness to reducing your risk of injury: 1 being the most useful to you and 7 the least useful to you. Note: drag and drop

______ Muscle Soreness (1)
______ Readiness to train (2)
______ Sleep Duration (3)
______ Sleep Quality (4)
______ Mood (5)
______ Fatigue (6)
______ Training Load (RPE x duration) (7)

11.　If the IRISweb Training Load Monitoring System had to be streamlined, what 3 measures would you keep?

☐　Muscle Soreness (1)
☐　Readiness to train (2)
☐　Sleep Duration (3)
☐　Sleep Quality (4)
☐　Mood (5)
☐　Fatigue (6)
☐　Training Load (RPE x duration) (7)

12.　Did you encounter any problems or difficulties using the IRISweb Training Load Monitoring System? Please give details.

_________________________________________________________

13.　If your club used the IRISweb Training Load Monitoring System again next season, what changes do you think should be made?

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
