# Peer review of "The Development and Evaluation of a Training Monitoring System for Amateur Rugby Union"

_applsci, doi:10.3390/app10217816_

Round 1

Reviewer 1 Report

General Comments:

The authors’ have undertaken a comprehensive process in the development and evaluation of a training monitoring system, that has potential application to amateur rugby and perhaps other team sports by guiding future implementation of such a system.  It is clear there are a number of barriers to the effectiveness of the TMS, which does bring into question the usefulness of the TMS that has been developed.  Nonetheless, it is helpful for researchers and practitioners to be aware of these barriers before trying to implement a TMS in future and the information presented in the manuscript has merit.

Specific Comments:

Introduction:  More information relating to the challenges experienced to training, monitoring training and the factors contributing to the injury incidence rate is relevant for the introduction.  The is particularly relevant given the issues you go on to discuss around players working in manual labour, or balancing evening shifts with early morning classes.  These are not issues that the professional game need to contend with and should be highlighted from the outset.

Methods & Materials: Acknowledging the volume of data you did collect, and through utilising different approaches, it is a shame that there is no comparison of your findings to objective data.  Considering this study is an extension of the IRIS, which presumably also collated injury incidence data for the 2019-20 season and multiple seasons beforehand, comparing the injury incidence data for the teams that participated in the TMS pre and post implementation would have been interesting.  If this is something that can be included it would strengthen your conclusions.

In section 2.2.1, you define each of the subjective pre-session measures, but did the players also receive information on how to interpret these terms as part of the questions?  For example, the question on soreness on the TMS screenshot does not specifically refer to muscle soreness unlike your in-text definition. Can you be certain that all players were recording the pre-session measures you intended?

Do you have any background information on the S&C coaches’ age and experience etc., as this would likely influence their ability to 1) use a smartphone interface to record and feedback data, and 2) their ability to interpret and utilise the data to benefit the player wellbeing/reduce injury risk.

Results: Section ‘2.5.1 Coach interview and survey’ implies there is a coach interview and coach survey, when it appears there is actually a coach interview and player survey.  Clarification here is required.

Table 1:

Line 2 – Should read ‘Were there any training monitoring practices in place prior to this season?’

Line 12 – Should read ‘How useful were the post session measures in terms of…’

Line 16 – Should read ‘What challenges or problems did you meet…’

Figure 4 – Do not join the mean data points when the data are such distinct categories. 

Section 3.1.1, Page 12, Line 3 (just below table 3) – Should read ‘All 6 S&C coaches…’

Discussion: It would be useful for you to acknowledge some of the points raised in the comments in the Materials and Methods section as part of your discussion.

Additionally, I recognise that you have made some suggestions throughout your discussion on how to overcome many of the challenges identified to using the TMS.  That said, it would be helpful for readers, and particularly for practitioners working in the amateur game wanting to make use of your TMS, to provide a summary of specific actions that should be taken to ensure future applications of the TMS are more successful and progress the work you have done here.

Conclusion: Line 8 – Should read ‘S&C coaches’

Author Response

SUMMARY OF REVISIONS

REVIEWER #1:

General Comments:

  • The authors’ have undertaken a comprehensive process in the development and evaluation of a training monitoring system, that has potential application to amateur rugby and perhaps other team sports by guiding future implementation of such a system.  It is clear there are a number of barriers to the effectiveness of the TMS, which does bring into question the usefulness of the TMS that has been developed.  Nonetheless, it is helpful for researchers and practitioners to be aware of these barriers before trying to implement a TMS in future and the information presented in the manuscript has merit.

Thank you for taking the time to review this study and we appreciate your critical yet very informative feedback. We have endeavoured to address each of the points raised by the reviewer. 

Specific Comments:

  • Introduction:  More information relating to the challenges experienced to training, monitoring training and the factors contributing to the injury incidence rate is relevant for the introduction.  The is particularly relevant given the issues you go on to discuss around players working in manual labour, or balancing evening shifts with early morning classes.  These are not issues that the professional game need to contend with and should be highlighted from the outset.

We sincerely appreciate you drawing our attention to this very important point. To address this we have added the following text to the introduction in order to highlight the disparities between the professional and amateur level as well as the unique challenges of the latter:

“Differences between professional and amateur rugby in terms of injury epidemiology, injury incidence rates, physical characteristics of players, and challenges to contend with have been highlighted in previous research [23]. The majority of the world’s rugby playing population however, compete at an amateur level. Despite this, there is a paucity of information in relation to monitoring training at this level and the inherent challenges this presents [23]. There is a need therefore, to examine the amateur game as its own distinct entity where staff and resources are reduced [23,43].”

We have also referred to the distinctions between the professional and amateur playing levels in the opening paragraph of the introduction which serves as a good preface to this important point:

“A successful training monitoring system (TMS) requires effective and consistent collection, analysis, and application of data that will allow adjustment of training prescription without overburdening the players [43,46]. This can be a particular challenge at the amateur level due to a lack of resources, time and financial compensation received by the S&C coaches [23].”

We feel these combined better introduce the reader to the concepts that are discussed in more detail throughout the other sections of the article. Thank you again for highlighting this.

  • Methods & Materials: Acknowledging the volume of data you did collect, and through utilising different approaches, it is a shame that there is no comparison of your findings to objective data.  Considering this study is an extension of the IRIS, which presumably also collated injury incidence data for the 2019-20 season and multiple seasons beforehand, comparing the injury incidence data for the teams that participated in the TMS pre and post implementation would have been interesting.  If this is something that can be included it would strengthen your conclusions.

Thank you for this suggestion. We agree that the comparison between the training monitoring data and the objective injury data would give another dimension to this research. However, the current paper has a number of key aims already including:

(1) Detailing the development of an online TMS for amateur rugby informed by existing literature and current S & C practices.

(2) Evaluating the effectiveness of the system from both the coach and player perspectives.

(3) Highlighting the player and coach challenges that can impact the effectiveness of a TMS.

(4) Providing informed recommendations that end users should take into consideration when devising and implementing future methods of training load monitoring .

We feel that the paper in its current format therefore has a wide scope to begin with and there is simply not the scope to add an extra variable (injury). We feel that the focus of the current article may be lost if we do and so should remain as offering of an effective and achievable method of monitoring training as well as highlighting barriers that need consideration. We do however hope to examine the training monitoring data and the injury data in a future article focussing solely on that aspect. We again thank you for your suggestion and hope our explanation is satisfactory.

  • In section 2.2.1, you define each of the subjective pre-session measures, but did the players also receive information on how to interpret these terms as part of the questions?  For example, the question on soreness on the TMS screenshot does not specifically refer to muscle soreness unlike your in-text definition. Can you be certain that all players were recording the pre-session measures you intended?

Thank you for bringing this to our attention. All the measures were defined and explained in detail in the introduction presentation, however this was not clearly outlined in the article. To add this clarity, we have now included the following text in the ‘Recruitment and introduction to the training monitoring system’ section of the methods and materials:

“The TMS was introduced to the players and coaches by means of a 20-minute presentation, given in person by the primary author or via video recording. This presentation outlined the purpose and potential benefits of their participation, the definition and explanation of each of the measures included and a demonstration of how to use the system.”

We hope this satisfies the reviewers points above.

  • Do you have any background information on the S&C coaches’ age and experience etc., as this would likely influence their ability to 1) use a smartphone interface to record and feedback data, and 2) their ability to interpret and utilise the data to benefit the player wellbeing/reduce injury risk.

Important point – thank you. We have now included the age and coaching experience of the 6 S&C coaches: (age = 30 ± 5 years, coaching experience = 5 ± 2 years): We feel the relatively young age of the coaches, combined with their presence at the introductory presentation, alleviated the issues outlined above.  

  • Results: Section ‘2.5.1 Coach interview and survey’ implies there is a coach interview and coach survey, when it appears there is actually a coach interview and player survey.  Clarification here is required.

To avoid confusion, we have now edited the headings in the methods and materials section and the result section to read “coach interview” rather than “coach interview and survey”. Thank you for bringing this to our attention.

Table 1:

  • Line 2 – Should read ‘Were there any training monitoring practices in place prior to this season?’

Revision made – thank you.

  • Line 12 – Should read ‘How useful were the post session measures in terms of…’

Revision made – thank you.

  • Line 16 – Should read ‘What challenges or problems did you meet…’

Revision made – thank you again.

  • Figure 4 – Do not join the mean data points when the data are such distinct categories. 

Valid comment – thank you. We have removed the line between the mean data points.

  • Section 3.1.1, Page 12, Line 3 (just below table 3) – Should read ‘All 6 S&C coaches…’

Revision made – thank you.

  • Discussion: It would be useful for you to acknowledge some of the points raised in the comments in the Materials and Methods section as part of your discussion.

As per our earlier response above, we feel there is just not the scope to examine injury in this particular article.

  • Additionally, I recognise that you have made some suggestions throughout your discussion on how to overcome many of the challenges identified to using the TMS.  That said, it would be helpful for readers, and particularly for practitioners working in the amateur game wanting to make use of your TMS, to provide a summary of specific actions that should be taken to ensure future applications of the TMS are more successful and progress the work you have done here.

This is an excellent point. To address it we have now extended the key findings section and separated it into two separate sections “What are the key findings?” and “What are the key practical applications for coaches?”. We hope by doing this we have now drawn attention to the specific actions that will help practitioners in monitoring training.

“What are the key findings?

  • Both the strength and conditioning (S&C) coaches and players alike value an online training monitoring system (TMS) but the greatest barriers to its effectiveness are lack of player compliance, data inconsistency, gathering match data, and external confounders outside of the sport.
  • Muscle soreness, fatigue, sleep duration and readiness to train (RTT) were highly rated as effective pre-session measures and are particularly useful at signalling red flags that prompt conversations between coaches and players.
  • Session rating of perceived exertion (sRPE) was a highly rated metric by S&C coaches and players for aiding training prescription and mitigating injury risk.

What are the key practical applications for coaches?

  • This study provides further support to the use of subjective measures as a method of monitoring training. Their ease of use, accessibility and low cost make them practically applicable in the amateur setting.
  • For players to engage with a TMS and give consistent data, they need regular feedback and evidence that the data are informing their training. This should aid in achieving consistent data from players and in turn improve the effectiveness of the TMS.
  • S&C coaches should be aware that the temporal robustness of sRPE may alleviate the difficulties around collecting post-match sRPE.
  • It is important that S&C coaches have the full support and assistance of all members of the coaching staff to ensure player compliance and to share the associated training monitoring and analysis demands.”

Conclusion: Line 8 – Should read ‘S&C coaches’

Revision made – thank you.

Thank you for taking the time to review this manuscript and we sincerely appreciate your thorough and critical feedback. We hope we have satisfied your points above and feel that the revisions suggested have greatly improved the revised manuscript.

Reviewer 2 Report

With substantial revisions, this paper may reach a publishable level. My comments/questions:

  1. The topic is interesting to readers but the question arises: did the data, obtained by the authors, confirm the hypothesis that online TMS is a useful and effective method of facilitating training prescription and design in an effort to reduce injury risk and enhance performance. What have we learned from this method that we didn’t already know? What is the added value of this method?
  2. What is the possibility that the method proposed by the authors is valid for the research aims? How this study result (data) correlate with the physical condition of the subjects? Does the methodology proposed by the authors explore the physiological state or the psychosomatic state of subjects?
  3. Although the authors provide some evidence of why this methodology is important to consumers, I am not convinced that it is enough. Why these objects of questions is chosen as variables and no other variables?
  4. Could we use this method to analyse athletes' physiological condition?
  5. The authors should explain the advantages of the proposed method compared to alternative physiological methods. What are the advantages and disadvantages of the proposed method?
  6. Is the proposed method reliable?
  7. It is not clear why the proposed method gives practical guidance. On what basis? What do you mean by practical settings?
  8. What are the methodological implications of the proposed method? Is this method useful for sports science?
  9. The conclusion of this review is that the article should be improved and research data have to be more clarity and precise.

Author Response

REVIEWER #2:

With substantial revisions, this paper may reach a publishable level. My comments/questions:

Thank you for taking the time to review this study and we appreciate your critical yet very informative feedback. We have endeavoured to address each of the points raised by the reviewer. 

  1. The topic is interesting to readers but the question arises: did the data, obtained by the authors, confirm the hypothesis that online TMS is a useful and effective method of facilitating training prescription and design in an effort to reduce injury risk and enhance performance. What have we learned from this method that we didn’t already know? What is the added value of this method?

Thank you for this comment. The purpose of this study was to:

(1) Detail the development of an online TMS for amateur rugby informed by existing literature and current S&C practices.

(2) Evaluate the effectiveness of the system from both the coach and player perspectives.

(3) Highlight the player and coach challenges that can impact the effectiveness of a TMS.

(4) Provide informed recommendations that end users should take into consideration when devising and implementing future methods of training monitoring.

We achieved the first purpose of the study by explicitly detailing the development, design and implementation of the TMS which will allow coaches working with teams to establish a similar system. In the methods and materials section, we also presented justification for each of the measures included in the TMs by referring to the current practices [23] and existing literature [7,10,14,18,23,24,28-30,35,40-42,44-46,48-51,53,54].

We achieved the second study purpose by comprehensively interviewing the 6 S&C coaches that used the system across a full season under the following key areas: coach and team profile, coach and player system use, system uptake and effectiveness, coach’s use of the data, challenges met and changes recommended going forward. The results of this study demonstrate that the coaches found the system to be moderate to highly effective in terms of both mitigating injury risk and enhancing performance.

We surveyed 72 players that used the system which consisted of questions that were aimed at gathering data regarding the overall experience, effectiveness, functionality, burden, and recommended changes to the TMS. The results showed that the players found it to be useful and easy to use.

This study highlights the challenges faced by coaches and players when using a TMS at the amateur level. For the coach, these challenges include lack of player compliance, data inconsistency, lack of support, addressing external confounders and getting data around match-day. For the players, the key challenges include receiving feedback on their data and seeing evidence of their data informing their training. These challenges negatively impacted the effectiveness of the system.

This study offers possible solutions to many of these challenges that we feel has the potential to further increase its effectives when used by coaches in the future. These solutions include:

(1) The player compliance – coach feedback loop whereby the coaches give regular feedback and evidence of data use and in turn players will have greater engagement resulting in more consistent data, thus increasing the effectiveness of the TMS.

(2) to alleviate the issue of lack of support, it is important that S&C coaches have the full support and assistance of all members of the coaching staff and club stakeholders to ensure optimal player compliance and to share the burden of monitoring TL.

(3) sRPE has been shown to be temporally robust for up to at least 24-hr post-exercise and a practical solution to alleviate the issue of collecting match data may be for practitioners to collect post-match sRPE the following day.

We appreciate the reviewer’s comments here and it is our plan to do future research that will take a quantitative approach in examining the training monitoring data and injury data, however it was not within the scope of this particular study nor was it the focus.

In terms of the value and what can be learned from this study, we strongly feel that the study highlights new insights that substantially progresses the research area: 

  • This study offers a comprehensive guide in developing TMS the is effective at the amateur level. There is a paucity of information in relation to monitoring training at the amateur level and its unique inherent challenges. This study begins to address this by offering a method of monitoring training that is attainable, scientifically grounded, and both S&C coaches and players find effective.

  • This is the first study of its kind to examine the perspectives of both coaches and players in relation to a TMS. The findings show that from the coaches’ perspective the TMS was useful and effective. This is supported by (1) the mean S&C coaches’ responses to its usefulness in terms of reducing injury risk being 3.5 / 5 and 3.3 / 5 for enhancing player performance and (2) the themes and representative quotes highlighted in Table 2. The findings of the study show that from the players’ perspective the system was effective across a range of measure (Figure 4), however it is fundamental to ensure that they receive regular feedback on their data and evidence of its use in training prescription and design.

  • This study highlights which measures the coaches and players found most effective and useful. Muscle soreness, fatigue, sleep duration and readiness to train (RTT) were highly rated as effective pre-session measures and are particularly useful at signalling red flags that prompt conversations between coaches and players. Also, sRPE was a highly rated metric by S&C coaches and players for aiding training prescription and mitigating injury risk.

  • This study highlights the unique challenges that can impact the effectiveness of a TMS at an amateur level. For the coach, these challenges include lack of player compliance, data inconsistency, lack of support, addressing external confounders and getting data around match-day. For the players, the key challenges include receiving feedback on their data and seeing evidence of their data informing their training.

  • This study offers possible solutions to many of these challenges including: (1) The player compliance – coach feedback loop whereby the coaches give regular feedback and evidence of data use and in turn players will have greater engagement resulting in more consistent data, thus increasing the effectiveness of the TMS. (2) to alleviate the issue of lack of support, it is important that S&C coaches have the full support and assistance of all members of the coaching staff and club stakeholders to ensure optimal player compliance and to share the burden of monitoring TL. (3) sRPE has been shown to be temporally robust for up to at least 24-hr post-exercise and a practical solution to alleviate the issue of collecting match data may be for practitioners to collect post-match sRPE the following day.

  • This study will aid practitioners working with teams at amateur level, in implementing a TMS despite limited resources.

Importantly, we feel this article offers practical approaches to improve the training monitoring practices of S&C coaches working with amateur teams and therefore will have real-world impact.  We hope this addresses the reviewer’s comment and we thank them once again.

  1. What is the possibility that the method proposed by the authors is valid for the research aims? How this study result (data) correlate with the physical condition of the subjects? Does the methodology proposed by the authors explore the physiological state or the psychosomatic state of subjects?

The aims of this study were:

(1) Detail the development of an online TMS for amateur rugby informed by existing literature and current S&C practices.

(2) Evaluate the effectiveness of the system from both the coach and player perspectives.

(3) Highlight the player and coach challenges that can impact the effectiveness of a TMS.

(4) Provide informed recommendations that end users should take into consideration when devising and implementing future methods of training monitoring.

Given these are the aims, it was not within the scope of this particular study to determine the validity of the system except to qualitatively determine its effectiveness through coach interviews and player surveys. It was also not within our aims to explore the physiological state or the psychosomatic state of subjects. In this article however, we have given justification for choosing the measure we included by referring to any available research that  supports its use, validity, reliability, as well as examples of when it was previously used:

sRPE: “This method of calculating TL has been shown to be valid and reliable across a wide variety of exercise modalities [18].”

Fatigue: “Accumulation of fatigue can result in a decrease in performance and an increase in injury risk and therefore monitoring fatigue is essential to understand the impact of TL [47].”

Muscle soreness: “The findings of McLean et al. [35] suggest that inappropriate TL can negatively impact on perceived muscle soreness resulting in a decrease in performance. It is therefore justifiable that the measure is commonly utilised as an important indicator of an athlete’s recovery state [48].”

We strongly feel that making changes to the article in terms of examining correlations with the physical condition, physiological state or the psychosomatic state of subjects  will cause it to lose its purpose and focus. We thank the reviewer again for their comments and hope we have given sufficient justification for not amending the article here.  

  1. Although the authors provide some evidence of why this methodology is important to consumers, I am not convinced that it is enough. Why these objects of questions is chosen as variables and no other variables?

Again, thank for this comment. The two central reasons for choosing the measures that were included in the TMS were:

  1. The survey of current practices highlighted that subjective measures are more applicable to amateur teams due to their ease of use, accessibility, and low cost.

  1. As outlined in the article, previous research supports and recommends the use of subjective measures:

“Subjective measures respond to training-induced changes in athlete wellbeing and may also be more sensitive and reliable than objective measures [10,14,44]. Therefore, it is recommended that athletes report their subjective wellbeing on a regular basis alongside other athlete monitoring practices [44].”

Importantly, for a system of this nature to be effective, its practicality is key (i.e. coaches need to get information to aid their practices, yet player burden has to be low enough as to not discourage them to use it regularly). Essentially, we chose the measures based on points 1 and 2 above and did not include other measures because it would over burden the players. One of the findings of this study is that of the 7 measures included in the TMS, there were 5 measures that were more highly rated by the coaches and players: sRPE, muscle soreness, fatigue, sleep duration and readiness to train. This has important connotations for coaches working in the field when developing their TMS whereby they may streamline the system further and thus increase its overall effectiveness. 

  1. Could we use this method to analyse athletes' physiological condition?

The purpose of this training monitoring system is to provide information to the coaches regarding an athletes’ preparedness for, and response to, various training stimuli. This may then provide practitioners with greater opportunity to prescribe and design training with the aim of maximising recovery and performance while simultaneously minimising risk of injury, illness and, health and wellbeing problems. More specific objective testing would be required to analyse an athlete’s physiological condition. This however, is impractical for coaches working with amateur teams that have limited time, limited access to such testing and limited financial resources to name just some barriers. Thank you and we hope this has addressed your question.

  1. The authors should explain the advantages of the proposed method compared to alternative physiological methods. What are the advantages and disadvantages of the proposed method?

The method of monitoring training that we have proposed in this article is:

  • cost effective
  • limits player burden
  • is deemed useful and effective by the coaches and players alike

These are all issues pertinent in the amateur game. We feel that these advantages result in this TMS being transferable to a practical setting (i.e. players playing amateur level Rugby and coaches working with amateur teams). We feel these advantages are well articulated throughout the article. However, to ensure the reader is better introduced to the differences between the professional and amateur game, and the associated challenges of working with an amateur team, we have added the following text to the introduction:

“Differences between professional and amateur rugby in terms of injury epidemiology, injury incidence rates, physical characteristics of players, and challenges to contend with have been highlighted in previous research [23]. The majority of the world’s rugby playing population however, compete at an amateur level. Despite this, there is a paucity of information in relation to monitoring training at this level and the inherent challenges this presents [23]. There is a need therefore, to examine the amateur game as its own distinct entity where staff and resources are reduced [23,43].”

We have also referred to the distinctions between the professional and amateur playing levels in the opening paragraph of the introduction which serves as a good preface to this important point:

“A successful training monitoring system (TMS) requires effective and consistent collection, analysis, and application of data that will allow adjustment of training prescription without overburdening the players [43,46]. This can be a particular challenge at the amateur level due to a lack of resources, time and financial compensation received by the S&C coaches [23].”

We feel these combined better introduce the reader to the concepts that are discussed in more detail throughout the other sections of the article. We hope this suitably addresses the reviewer’s comment and we thank them once again.

  1. Is the proposed method reliable?

Typically, qualitative research cannot verify reliability of a system of this nature.  In the introduction of this article however, we have given support for the use and reliability of subjective measures of training:

“Subjective measures respond to training-induced changes in athlete wellbeing and may also be more sensitive and reliable than objective measures [10,14,44]. Therefore, it is recommended that athletes report their subjective wellbeing on a regular basis alongside other athlete monitoring practices [44].”

Additionally, when describing the subjective measures used in the TMS in the methods and materials section, we have referred to any available research that  supports its use, validity, reliability, as well as examples of when it was previously used.

sRPE: “This method of calculating TL has been shown to be valid and reliable across a wide variety of exercise modalities [18].”

Fatigue: “Accumulation of fatigue can result in a decrease in performance and an increase in injury risk and therefore monitoring fatigue is essential to understand the impact of TL [47].”

Muscle soreness: “The findings of McLean et al. [35] suggest that inappropriate TL can negatively impact on perceived muscle soreness resulting in a decrease in performance. It is therefore justifiable that the measure is commonly utilised as an important indicator of an athlete’s recovery state [48].”

In the methods and materials section, we also offer support for the use of the 1 – 5 Likert scale to assess the 6 pre-session measures:

“A five-point Likert scale has been used to assess subjective measures of well-being in several previous articles [35,40,41,51].”

Research regarding the reliability of these measures is limited but, importantly, what is evident from our study is that the 6 S&C coaches that were interviewed and the 72 players that completed the survey (all who used the TMS for a full season) found it to be useful  and effective.

  1. It is not clear why the proposed method gives practical guidance. On what basis? What do you mean by practical settings?

This is an excellent point. Based on this, and the comments of another reviewer, we have now extended the Key Findings section and separated it into two separate sections “What are the key findings?” and “What are the key practical applications for coaches?”:

“What are the key findings?

  • Both the strength and conditioning (S&C) coaches and players alike value an online training monitoring system (TMS) but the greatest barriers to its effectiveness are lack of player compliance, data inconsistency, gathering match data, and external confounders outside of the sport.
  • Muscle soreness, fatigue, sleep duration and readiness to train (RTT) were highly rated as effective pre-session measures and are particularly useful at signalling red flags that prompt conversations between coaches and players.
  • Session rating of perceived exertion (sRPE) was a highly rated metric by S&C coaches and players for aiding training prescription and mitigating injury risk.

What are the key practical applications for coaches?

  • This study provides further support to the use of subjective measures as a method of monitoring training. Their ease of use, accessibility and low cost make them practically applicable in the amateur setting.
  • For players to engage with a TMS and give consistent data, they need regular feedback and evidence that the data are informing their training. This should aid in achieving consistent data from players and in turn improve the effectiveness of the TMS.
  • S&C coaches should be aware that the temporal robustness of sRPE may alleviate the difficulties around collecting post-match sRPE.
  • It is important that S&C coaches have the full support and assistance of all members of the coaching staff to ensure player compliance and to share the associated training monitoring and analysis demands.”

We hope by doing this we have now drawn attention to the specific actions that will help practitioners in monitoring training. The practical setting in this case being coaches working with amateur Rugby Union teams. We feel this has substantially improved the article.

  1. What are the methodological implications of the proposed method? Is this method useful for sports science?

We strongly feel this TMS is useful for sports science and has potential longer term benefits if used in conjunction with comprehensive injury and illness monitoring systems.  There is a paucity of information in relation to monitoring training at the amateur level and associated inherent challenges. This study begins to address this by offering a method of monitoring training that is attainable, scientifically grounded, and both S&C coaches and players find moderate to highly effective. Additionally, it establishes pitfalls to avoid that will be useful for both coaches working in the field and researchers intending to examine training using a monitoring system.

The potential benefits of the TMS are highlighted throughout the article , for instance:

“An online TMS can provide information that triggers these conversations, which in turn has the potential to support player performance and mitigate injury risk [37]. This is evident from the representative S&C coaches’ quotes in Table 2, for example, “That is massively beneficial. It’s a great conversation starter.” (Coach 6).”

“Practically, pre-session measures can prompt conversations between coaches and players while sRPE can inform S&C coaches’ training prescription and design. These measures combined, can potentially promote a reduction in injury risk and ultimately support player performance. There are several challenges to monitoring training in the amateur setting that will limit its success and effectiveness but potential resolutions to these have been outlined.”

Furthermore, the addition of the “What are the key practical applications for coaches?” within the Key Findings section in response to the reviewer’s point 7, now draws attention to the specific actions that will help practitioners in monitoring training.

  1. The conclusion of this review is that the article should be improved and research data have to be more clarity and precise.

We would like to thank the reviewer for their critical review of our manuscript. We have carefully considered and addressed all the points raised. We feel this has enhanced the clarity and precision of the paper.

Round 2

Reviewer 2 Report

Thank you for authors that they clearly explained and justified some unclear issues, and corrected the manuscript according to our review.